

# Evaluation of the Transport Matrix Method for simulation of ocean biogeochemical tracers

Karin F. Kvale[1], Samar Khatiwala[2], Heiner Dietze[1], Iris Kriest[1], and Andreas Oschlies[1]

[1]GEOMAR Helmholtz-Zentrum für Ozeanforschung Kiel, Düsternbrooker Weg 20, D-24105 Kiel, Germany
[2]Department of Earth Sciences, University of Oxford, South Parks Road, Oxford OX1 3AN, UK

*Correspondence to:* K. Kvale (kkvale@geomar.de)

**Abstract.** Conventional integration of earth system and ocean models can accrue considerable computational expenses, particularly for marine biogeochemical applications. "Offline" numerical schemes in which only the biogeochemical tracers are time-stepped and transported using a pre-computed circulation field can substantially reduce the burden and are thus an attractive alternative. One such scheme is the "transport matrix method" (TMM), which represents tracer transport as a sequence of
5 sparse matrix-vector products that can be performed efficiently on distributed-memory computers. While the TMM has been used for a variety of geochemical and biogeochemical studies, to date the resulting solutions have not been comprehensively assessed against their "online" counterparts. Here, we present a detailed comparison of the two. It is based on simulations of the state-of-the-art biogeochemical sub-model embedded within the widely-used University of Victoria Earth System Climate Model (UVic ESCM). Transport matrices were extracted for an equilibrium run of the physical model and subsequently used
to integrate the biogeochemical model offline to equilibrium. The identical biogeochemical model was also run online. Our simulations show that offline integration introduces some bias to biogeochemical quantities through the omission of the polar filtering used in UVic ESCM, and in the offline application of time-dependent forcing fields, with high latitudes showing the largest differences with respect to the online model. Differences in other regions and in the seasonality of nutrients and phytoplankton distributions are found to be relatively minor, giving confidence that the TMM is a reliable tool for offline integration
of complex biogeochemical models. Moreover, while UVic ESCM is a serial code, the TMM can be run on a parallel machine with no change to the underlying biogeochemical code, thus providing orders of magnitude speed-up over the online model.

## 1 Introduction

The transport matrix method (TMM) (Khatiwala et al., 2005; Khatiwala, 2007) is a numerical scheme for efficient simulation of ocean biological and chemical tracers. It is based on the idea that the advective-diffusive transport of a passive tracer is a lin-
20 ear operator which, when spatially discretised, can be generically represented as a sparse matrix. Time-stepping of such tracers is thus reduced to a sequence of sparse matrix-vector multiplications, operations that can be carried out efficiently on modern, distributed-memory computers. While conventional ocean general circulation models (OGCMs) do not typically represent transport in this manner, for many GCMs is it is possible to extract the corresponding matrix representation of the OGCM's tracer transport scheme, including sub-grid scale parameterizations, by "probing" it with patterns of 1's and 0's (Khatiwala





et al., 2005). The transport matrix approach is also amenable to direct computation of equilibrium solutions, including periodically (seasonally) repeating ones (Khatiwala, 2008). This is especially useful for "spin-up" of complex biogeochemical models, which requires several thousand year-long integrations.

The TMM has been applied to a wide range of problems, including: simulating anthropogenic carbon uptake and radiocarbon by the ocean (Graven et al., 2012); simulating noble gases to improve the parameterization of air-sea gas transfer (Nicholson et al., 2011; Liang et al., 2013) and investigate ocean ventilation (Nicholson et al., 2016); studying ocean proxy (Siberlin and Wunsch, 2011) and radiocarbon (Koeve et al., 2015) time scales; investigating the mechanisms controlling nutrient ratios (Weber and Deutsch, 2010); modeling the cycling of particle reactive geochemical tracers (Jones et al., 2008; Siddall et al., 2008; Vance et al., 2017); estimating respiration in the ocean from oxygen utilization rates (Duteil et al., 2013; Koeve and Kähler, 2016); demonstrating the utility of atmospheric potential oxygen measurements to constrain ocean heat transport (Resplandy et al., 2016); modeling the ocean's $CaCO_3$ (Koeve et al., 2014) and nitrogen (Kriest and Oschlies, 2015) cycles; studying the impact of the Southern Ocean on global ocean oxygen (Keller et al., 2016); estimating the flux of organic matter (Wilson et al., 2015); and biogeochemical parameter sensitivity (Khatiwala, 2007; Kriest et al., 2010, 2012) and optimization (Priess et al., 2013b, a; Kriest et al., 2017).

Despite these varied applications, a comprehensive evaluation of the TMM viz a vis results produced by the corresponding online model has not yet appeared in the published literature. Khatiwala et al. (2005) and Khatiwala (2007) provide a limited comparison between online and offline TMM simulations of a simple passive tracer and biogeochemical model using transport matrices extracted from the MITgcm (Marshall et al., 1997). For problems that use information gleaned from offline simulations to inform online simulations it is especially important that the offline simulations faithfully reproduce the online ones. Here, we provide the first such comprehensive assessment of the TMM based on version 2.9 of the University of Victoria Earth System Climate Model (UVic ESCM; Weaver et al., 2001; Eby et al., 2009), a coarse-resolution (1.8° × 3.6° × 19 ocean depth layers) ocean-atmosphere-biosphere-cryosphere-geosphere model. The marine biogeochemistry has increased in complexity since Eby et al. (2009), with the addition of iron limitation and revisions to zooplankton grazing (Keller et al., 2012), and subsequent bug fixes and minor updates. We describe the procedure to extract TMs from the UVic ESCM ocean model and couple the biogeochemical model to the TMM framework. An equilibrium simulation of the biogeochemical model with the TMM is then compared with a conventional online simulation. We discuss some of the compromises needed and their impact on the offline simulations.

## 2 Methods

### 2.1 Transport matrix extraction in UVic ESCM

The TMM relies on the fact that the underlying partial differential equation for transport of a passive tracer is linear with respect to advection and diffusion. If the discrete (numerical) implementation is also linear, the tracer time-stepping equation



can be generally written in matrix form as (Khatiwala, 2007):

$$\mathbf{c}^{n+1} = \mathbf{A}_\mathrm{i}(\mathbf{A}_\mathrm{e}\mathbf{c}^n + \mathbf{q}^n), \tag{1}$$

where $n$ is the time step, $\mathbf{c}$ is a vector of tracer concentrations (the 3-d model grid mapped onto a vector); $\mathbf{A}_\mathrm{e}$ is the "explicit" transport matrix representing horizontal advection-diffusion (and, commonly, vertical advection) that is generally time-stepped using an explicit, forward-in-time scheme; $\mathbf{A}_\mathrm{i}$ is the "implicit" transport matrix representing vertical diffusion (and, less commonly, vertical advection) that is typically time-stepped using an implicit method; and $\mathbf{q}$ is a vector representing the sources and sinks of the tracer. Note that $\mathbf{A}_\mathrm{i}$ can be thought of as the inverse of the tridiagonal matrix arising in the implicit solution of the diffusion equation.

The procedure for extracting the TMs is described in detail in Khatiwala (2007). In essence it involves: (1) initializing the tracer field at the start of each time step with a "1" at a single grid point and "0" everywhere else; (2) computing the explicit tracer tendency, which in effect gives the corresponding column of the explicit matrix $\mathbf{A}_\mathrm{e}$; (3) resetting the tracer field to its initial pattern; and (4) applying implicit vertical diffusion, the result of which is the corresponding column of $\mathbf{A}_\mathrm{i}$. The TMs are generally averaged over a number of time steps to give a time-mean matrix representation of the transport. In practice, the extraction can be considerably sped up by noting that many columns of the TMs are "structurally independent" since tracer spreads only a finite (known) distance in a single time step. Multiple columns of $\mathbf{A}_\mathrm{e}$ and $\mathbf{A}_\mathrm{i}$ can thus be simultaneously computed by a judicious choice of the pattern of 1's and 0's. Such an optimal set of "basis functions" can be computed by combining knowledge of the model bathymetry and stencil of the advection-diffusion scheme with graph coloring methods (Curtis et al., 1974; Coleman and Moré, 1983).

It is quite straightforward to implement the above procedure in an ocean model with only minor modifications to the code, and we have done so with UVic ESCM. However, in practice a number of complications can arise. For example, ocean models sometimes use a nonlinear advection scheme so as to avoid over- and undershoots arising from sharp gradients in the tracer field. In fact, UVic ESCM uses such a scheme, "flux-corrected transport" (FCT; Weaver and Eby (1997)), by default. To reconcile this with the linearity requirement of the TMM we instead use a $3^\mathrm{rd}$ order, upwind-biased scheme (Holland et al., 1998; Griffies et al., 2008) (hereafter "UW3") that significantly reduces potential over/undershoots but is also much less diffusive than conventional $1^\mathrm{st}$ order upwind. This scheme has already been implemented in UVic ESCM but switching it on caused the strength of the meridional overturning circulation (MOC) in the model to decrease significantly. To obtain a MOC strength consistent with observations we increased the vertical diffusion coefficient from 0.15 cm$^2$ s$^{-1}$ to 0.43 cm$^2$ s$^{-1}$. However, it is important to note that other parameters in the model, such as the atmospheric moisture diffusivity field, also have an impact on ocean circulation. These parameters were tuned for the FCT scheme but were left unchanged when switching to the UW3 scheme.

A second complication is from the time-stepping scheme, which in UVic ESCM is leapfrog. The explicit horizontal advection and diffusion terms are also sometimes staggered with respect to each other for stability. While both can be replicated offline, in order to use a common scheme for all ocean models from which TMs have been extracted (e.g., MITgcm variously uses





Adams-Bashforth, direct space-time discretization and other schemes), we combine horizontal advection and diffusion into a single explicit transport matrix, $\mathbf{A}_e$ which is time-stepped with a simple Euler method.

Lastly, UVic ESCM applies Fourier filtering in the zonal direction at high latitudes to remove grid-scale noise. This polar filtering has the effect of making the TMs dense. We therefore turn off polar filtering for the passive tracers used to extract the

5 TMs. The numerical treatment of temperature and salinity by the model is not altered.

Using the linear UW3 advection scheme, the coupled physical-biogeochemical model was spun-up to equilibrium for 13,000 years with a pre-industrial atmospheric $CO_2$ concentration of 277.4 ppm. Monthly mean TMs from the equilibrium state were then extracted by running the model for one additional year. For comparison, we also carried out a similar spin-up of the physical and biogeochemical model using the default FCT advection scheme.

## 2.2 Interfacing the UVic ESCM biogeochemical model to the TMM

Once the TMs have been extracted, biogeochemical tracers can be simulated offline via Eq. 1 using any convenient software. However, to facilitate use of the TMM for global simulations of complex biogeochemical models, Eq. 1 has been implemented using the open source PETSc scientific library from Argonne National Laboratories (Balay et al., 2003). PETSc is a suite of data structures and routines developed for large scale linear and nonlinear problems. In particular, it provides state-of-the-art,

distributed-memory routines for operating on vectors and sparse matrices. The TMM driver code implementing Eq. 1 is freely available from https://github.com/samarkhatiwala/tmm. The code can be run without modification on most computers, serial or parallel, with details of the parallelism handled by PETSc and "hidden" from the end user. Additionally, the interface is implemented in a generic fashion, with the potential to "mix-and-match" circulation (transport matrices) and biogeochemical models.

To apply the TMM code to a particular biogeochemical problem essentially requires providing a routine that takes as input vertical "profiles" of tracer concentrations at a horizontal location at the current time step (along with any external variables such as temperature and wind speed at that location), and returns profiles of the biogeochemical tendency term, $\mathbf{q}$. In practice, coupling an existing biogeochemical model such as the one in UVic ESCM to the TMM framework involves writing a "wrapper" routine that serves as an interface between the TMM driver and the biogeochemical code. While the specific imple-

mentation of the wrapper will depend on the details of the biogeochemical model, to couple the UVic ESCM biogeochemical model required only minimal changes to the original code. Most of those changes were necessary so as to make available the full set of diagnostics accumulated by UVic ESCM. See Sec. 5 for information on where to download the code from.

## 3 Results and Discussion

### 3.1 Comparison of 3$^{rd}$-order upwind and flux-corrected transport online runs

We begin by discussing the changes due to switching from UVic ESCM's default, nonlinear flux-corrected transport scheme (UVIC_FCT) to the 3$^{rd}$-order linear advection scheme (UVIC_UW3). As noted above, only the vertical diffusivity was in-





creased when switching to the UW3 scheme to bring the MOC strength back closer to observations; all other parameters were left at values previously tuned to the default FCT scheme.

Fig. 1 compares the global, Atlantic, and Indo-Pacific meridional overturning circulation of the model for the two schemes. Antarctic Bottom Water (AABW) formation in the Southern Ocean occurs with an anti-clockwise streamfunction of 2 Sv poleward of a Deacon cell reaching 32 Sv near the surface in both cases. AABW extends northward into the Atlantic with an anti-clockwise streamfunction of up to 2 Sv below 3000 meters depth, and up to 6 Sv in the deep Indo-Pacific, with only minor differences in isolines between the two schemes. North Atlantic Deep Water (NADW) overlies AABW and reaches a maximum of 16 Sv around 30°N and 1000 meters depth. Overturning is slightly stronger in this region in UVIC_UW3 than in UVIC_FCT, with the zone within the 14 Sv isoline extending from about 5 to 40°N in the former, as opposed to roughly 20 to 40°N in the latter. This is mirrored by the distribution of natural radiocarbon (Fig. 2), which shows a larger North Atlantic contribution in UVIC_UW3 leading to generally younger deep ocean radiocarbon ages in this configuration relative to UVIC_FCT.

Sea surface temperature and zonally-averaged sections of potential temperature in each ocean basin show few differences between UVIC_UW3 and UVIC_FCT (Fig. 3). The most notable one is in the abyssal Atlantic, where the 4°C isotherm extends 1000 meters deeper and the 0°C isotherm extends 20° farther north in UVIC_UW3, consistent with a stronger NADW formation and more energetic AABW in this simulation. Both cases show a 1-2°C cold bias in the abyssal Atlantic and Pacific basins, and a warm bias of up to 5°C in the tropics, with respect to an observational climatology (World Ocean Atlas; Locarnini et al. 2010). Similarly, salinity differences between the two schemes are most notable in the Atlantic basin (Fig. 4). UVIC_UW3 AABW is 0.5 psu fresher than UVIC_FCT, though the northern Atlantic salinity is more similar. Surface salinity in both cases is generally too low relative to observations (Antonov et al., 2010), while surface and deep north Atlantic salinity is 0.5-1.0 psu too high.

The Taylor diagrams shown in Fig. 5 highlight the impact of the advection scheme on biogeochemistry. As in the physical model, the biogeochemical model parameters were previously tuned for the FCT advection scheme; no additional tuning was performed for the UW3 scheme. We therefore expect UVIC_FCT to be in closer agreement to observations than UVIC_UW3, although, generally, differences between UVIC_FCT and UVIC_UW3 biogeochemistry are much smaller than their mismatch with observations. The largest differences are in nitrate and alkalinity, both of which are sensitive to small differences in oxygen.

## 3.2 Comparison of online and offline simulations

### 3.2.1 Mean state

We next compare the annual-mean state of the online (UVIC_UW3) and offline (UVIC_TM) simulations. The surface dissolved inorganic carbon (DIC) distribution (Fig. 6) typically differs by less than 5 $\mu$mol kg$^{-3}$, although isolated grid points in the Arctic are up to 20 $\mu$mol kg$^{-3}$ lower in UVIC_TM. In the low latitudes there are grid points with differences of up to 10 $\mu$mol kg$^{-3}$ due to minor differences in the phytoplankton distribution between UVIC_TM and UVIC_UW3 (Fig. 7). Diazotroph populations in the Indian and western Pacific are more concentrated in UVIC_TM, with annual average concentra-



tions exceeding 0.3 mmol N m$^{-3}$ at some locations. Differences in the diazotroph distribution can influence DIC and alkalinity (Fig. 8) because diazotrophs do not contribute to carbonate export in the model. Diazotrophs are disproportionately sensitive to low oxygen levels as oxygen deficiency can trigger denitrification, which increases their relative fitness over the general phytoplankton type in our model. Small spatial and temporal differences in suboxic conditions in the on- and offline models

can therefore affect diazotrophs in this region where oxygen levels are already fairly low (Fig. 9). Thus the distribution of carbon and alkalinity is slightly affected. Note, though, that online-offline differences in these quantities are restricted to the surface and do not extend into the deep ocean.

Similar to DIC, differences in surface alkalinity values rarely exceed 5 $\mu$mol kg$^{-3}$ (Fig. 8). Sub-surface alkalinity values in the Arctic are up to 50 $\mu$mol kg$^{-3}$ too high in UVIC_TM, likely because of the absence of polar filtering. Zonally-averaged

sections show that alkalinity is otherwise in close agreement between on- and offline integrations.

Phosphate and nitrate distributions likewise show negligible differences between models with the exception of polar regions and the water masses closely associated with those regions (Figs. 10 and 11). The UVIC_TM phosphate concentrations are up to 0.2 mmol m$^{-3}$ higher in the deep sub-polar Atlantic and abyssal Southern Ocean, while UVIC_TM nitrate concentrations are up to 5 mmol m$^{-3}$ higher than UVIC_UW3 in the abyssal Southern Ocean, but up to 5 mmol m$^{-3}$ lower in the deep

sub-polar Atlantic. The absence of polar filtering in the offline model is the most plausible explanation for these differences, all of which are small relative to discrepancies between the models and observations (see below).

Oxygen distributions in UVIC_TM are up to 50 mmol m$^{-3}$ higher in the deep sub-polar Atlantic and in parts of the surface Arctic compared with UVIC_UW3 (Fig.9). Aside from the minor differences at lower latitudes mentioned earlier, these are the only notable mismatches between the off- and online integrations for oxygen.

The above results are summarized in a spatially-integrated manner via the Taylor plots and associated tables shown in Fig. 12. Evidently, the root mean square (RMS) difference between the offline and online runs for most variables is quite small. To put those differences into perspective, also tabulated is the RMS difference between the online simulation and observations, which is seen to be over 10 times larger than the corresponding RMS difference between offline and online runs.

### 3.2.2  Seasonal cycle

In addition to the time-mean state it is important for any offline approach to capture the seasonal cycle. Fig. 13 shows that this is indeed the case for the zonally-averaged biomass at the surface. The relative error resulting from small differences in small concentrations can be large, so this comparison is limited to noting that the relative error in biomass generally increases towards the poles, as would be expected based on the discrepancies in annually-averaged biogeochemical quantities described earlier.

The timing of seasonal maxima also appear well-aligned between offline and online runs when examined at several locations (Fig. 14). The North Pacific, North Atlantic and Southern Ocean locations in particular display synchronous seasonal timing in nutrients and phytoplankton, although there are small differences in the amplitude at a few locations. This may in part be due to differences in how the offline and online models are forced. Similar to other offline schemes, in the TMM time-averaged circulation and forcing fields (for example, insolation, dust (iron) flux, sea ice and wind speed) are linearly interpolated in



time to the current time step before being applied. This averaging, the degree of which (monthly in these experiments) is arbitrary, has the potential to introduce biases, especially in rapidly-varying variables such as phytoplankton that are sensitive to nonlinearities in the population equations.

### 3.3 Computational performance of the TMM versus online model

5 The UVic ESCM model is a serial code and thus unable to exploit more than one computational core. With the biogeochemical component switched on the model throughput on a typical Linux machine is about ∼250 model years per day. A 5000-year spin-up of the online biogeochemical model thus takes ∼3 weeks. On the other hand, the PETSc-based TMM version can run in parallel, even though the underlying biogeochemical code is identical. While we have not carried out a detailed scaling analysis of the TMM version's performance, a similar 5000-year spin-up can be accomplished in 3.8 hours with 256 cores 10 on NCAR's Yellowstone IBM iDataPlex cluster, and in 5.2 hours with 160 cores on GEOMAR/Kiel University's NEC HPC machine.

### 4 Conclusions

This study investigates the extent to which the transport matrix method, a scheme for offline simulation of ocean biogeochemical tracers, can reproduce the corresponding online model, specifically an NPZD-type biogeochemical model embedded into 15 UVic ESCM. While the focus is on a detailed comparison of the offline run with the online one, we also describe the mechanics of extracting transport matrices from UVic ESCM and coupling it's biogeochemical model to the TMM framework. As the steps required for both aspects are quite general, this may be useful for researchers interested in applying the TMM to other ocean and biogeochemical models.

We show that the TMM version captures reasonably well both the time-mean state and seasonal cycle of biogeochemical 20 tracers of the online model. Small discrepancies arise at high latitudes due to the absence of polar filtering in the offline model. Differences also arise from the time averaging (monthly in these experiments) of forcing fields and the circulation embedded in the transport matrices, although the degree of averaging can be varied to suit the situation. Phytoplankton are especially sensitive to this time averaging, with the impact on most biogeochemical tracers being limited to the surface ocean. These differences are generally much smaller than biases in the models with respect to observations, and should be put in 25 the context of the roughly hundred-fold improvement in wall clock time from using the TMM. While UVic ESCM is a serial model, the TMM version can be run in parallel without any modifications to the underlying code. Simulations performed with the TMM are thus orders of magnitude faster making it possible to routinely perform long spin-ups of UVic ESCM in a few hours rather than weeks. A recent study (Seferian et al., 2016) highlighting the importance of adequate spin-ups suggests that this could be beneficial even for earth system models that are already parallelized, especially with the advent of 30 many-core hardware architectures, such as general purpose graphics processing units (GPGPUs) to which the TMM has been recently ported (Siewertsen et al., 2013). Moreover, the speed-up opens up the possibility of systematically testing different parameterizations in complex, global biogeochemical models, or even optimizing such models against data as has been recently



accomplished for a slightly simpler model by Kriest et al. (2017). While the results presented here are for a particular model, they should be broadly applicable to other global models of similar complexity.

## 5 Code availability

The specific version of the offline biogeochemical model code, transport matrices and other data used for the simulations

described in this paper are available from https://thredds.geomar.de/thredds/catalog/open_access/Kvale_etal_2017_GMDD/ catalog.html or http://kelvin.earth.ox.ac.uk/spk/Research/TMM/Kvale_etal_2017_GMDD.tar.gz. (The most recent version of the code and transport matrices from various circulation models are distributed from https://github.com/samarkhatiwala/tmm.) The archive also contains model output and scripts used to create the figures in this manuscript.

*Author contributions.* S. Khatiwala and K. Kvale wrote the paper (with comments by H. Dietze, I. Kriest, and A. Oschlies), implemented

the matrix interface and extraction, and performed the simulations. H. Dietze re-tuned the online model to use a linear advection scheme and assisted with interpretation. A. Oschlies, I. Kriest and S. Khatiwala conceived the project.

*Acknowledgements.* This work is a contribution to DFG-supported project SFB754 and to the research platforms of DFG cluster of excellence The Future Ocean. Khatiwala was supported in part by US NSF grant OCE 12-34971 and UK NERC grant NE/K015613/1. Computing resources (ark:/85065/d7wd3xhc) were provided by the Climate Simulation Laboratory at NCAR's Computational and Information Systems

Laboratory, sponsored by the National Science Foundation and other agencies. We also gratefully acknowledge computer resources made available at Kiel University, technical consultation with M. Eby, and figure scripts by A. Schmittner.



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





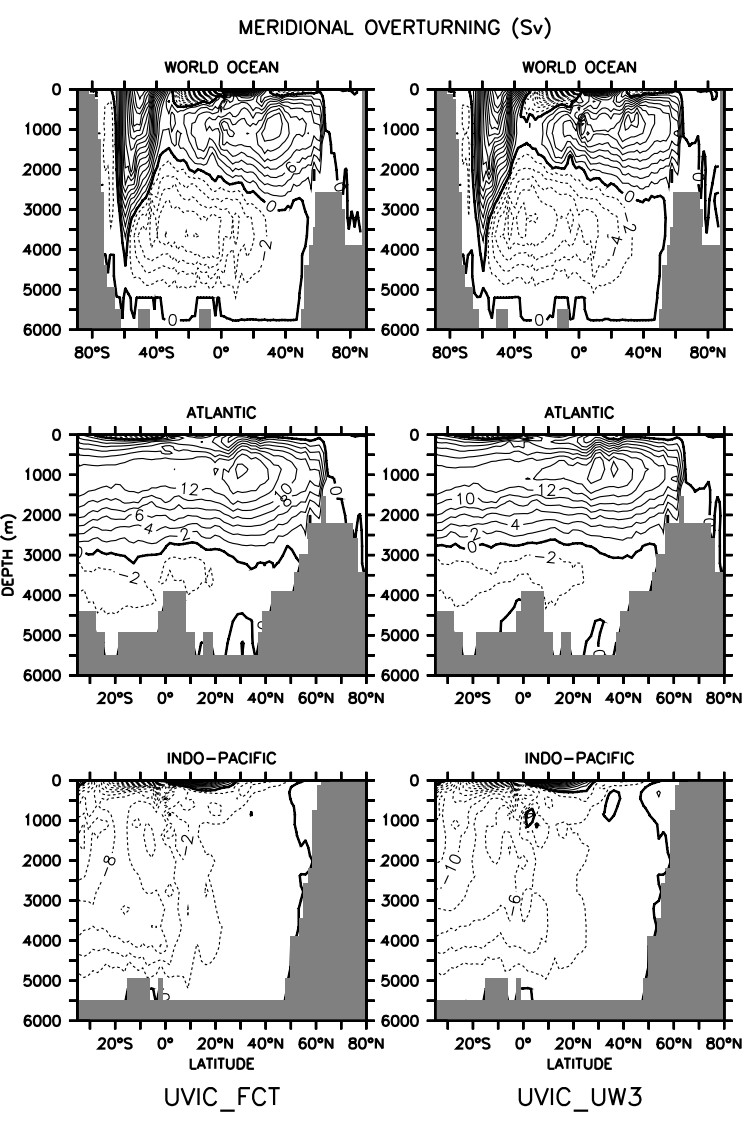

**Figure 1.** Meridional overturning in flux-corrected transport (UVIC_FCT) and third-order upwind advection (UVIC_UW3) online models. Positive (negative) values indicate clockwise (anti-clockwise) circulation.





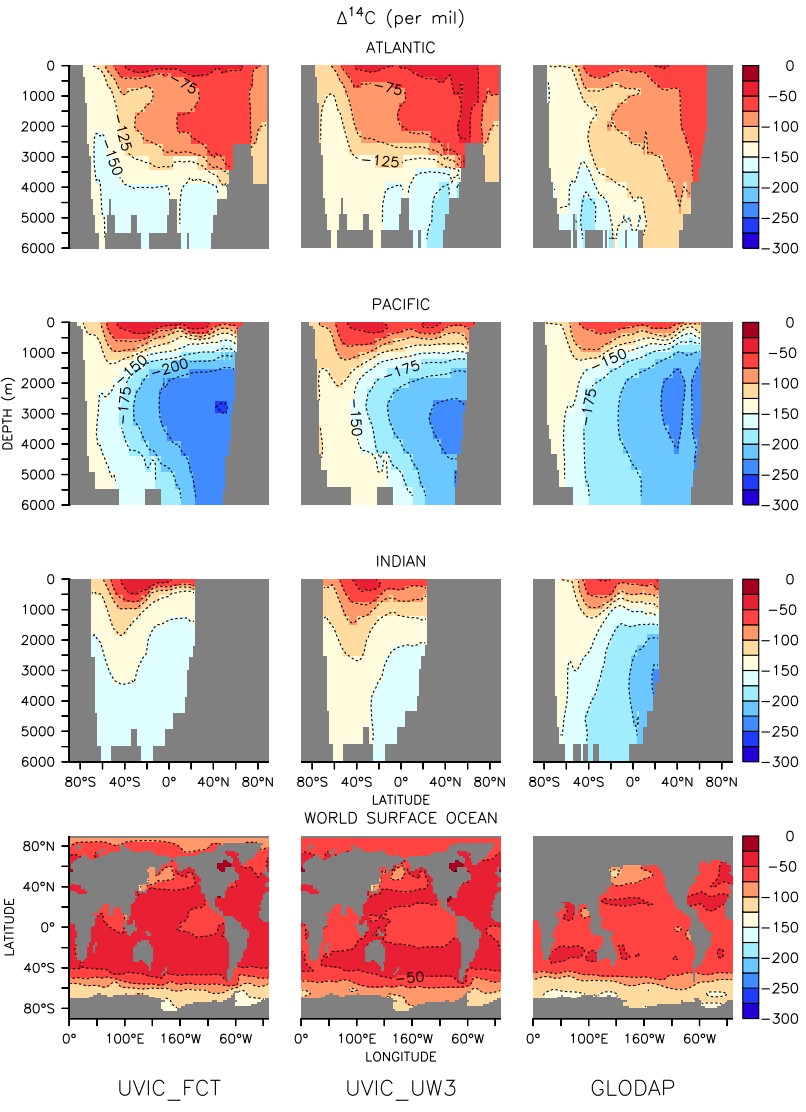

**Figure 2.** Background radiocarbon in online models UVIC_FCT and UVIC_UW3 compared with gridded observations from the GLODAP dataset (Key et al., 2004).





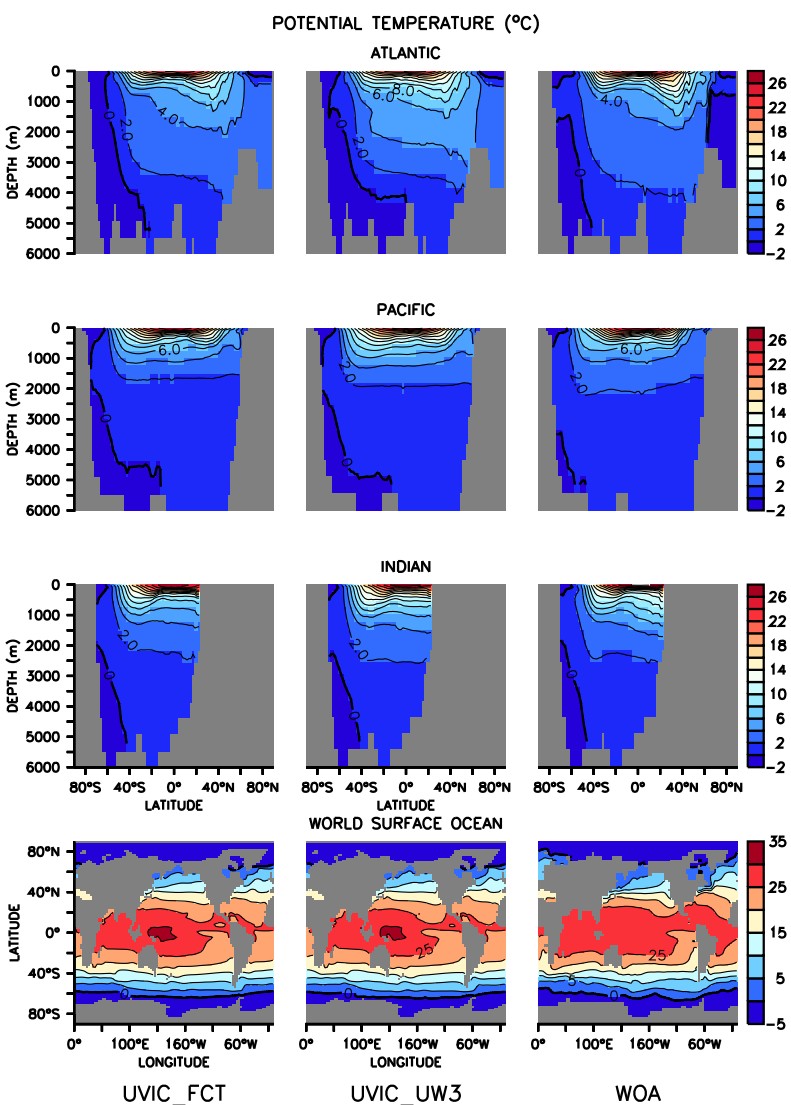

**Figure 3.** Potential temperature in online models UVIC_FCT and UVIC_UW3 compared with the World Ocean Atlas climatology (Locarnini et al., 2010).





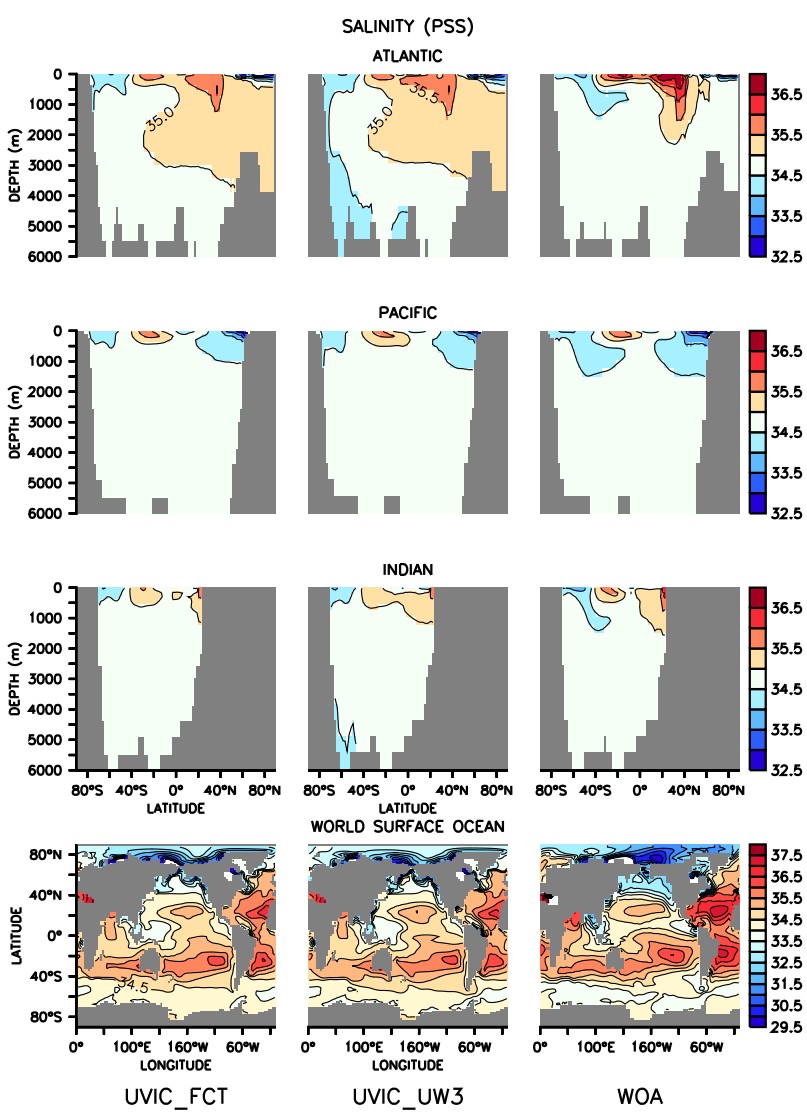

**Figure 4.** Salinity in online models UVIC_FCT and UVIC_UW3 compared with the World Ocean Atlas climatology (Antonov et al., 2010).





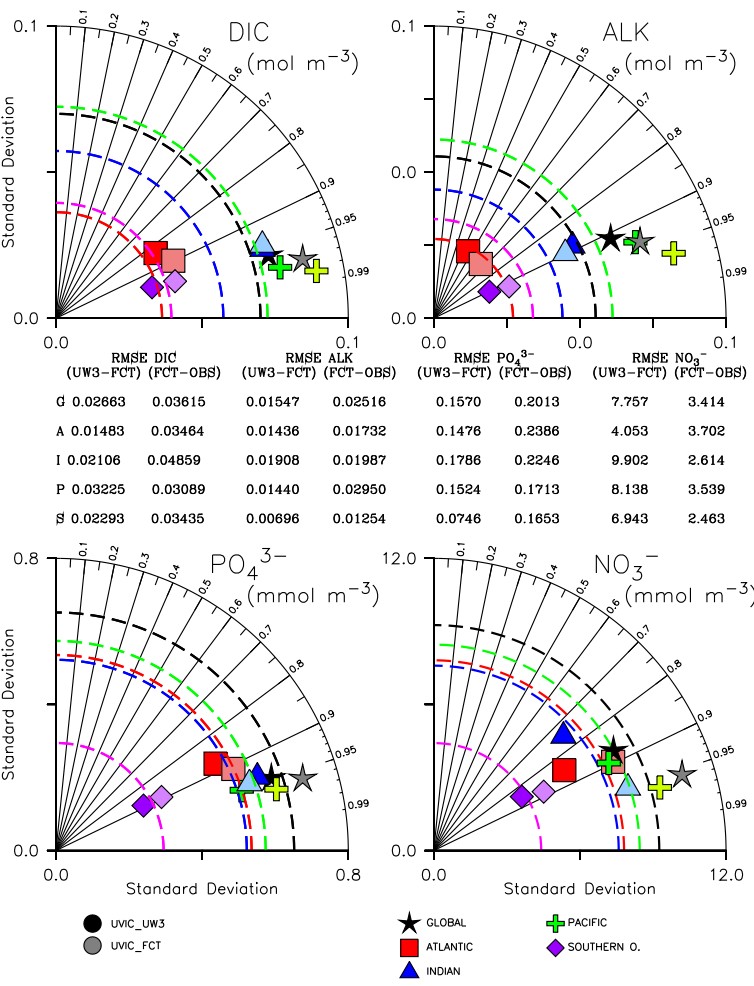

**Figure 5.** Taylor diagrams for various simulated biogeochemical quantities in comparison to observations (World Ocean Atlas climatology (Garcia et al., 2010a, b)) showing the impact of advection scheme. Different symbols and colors correspond to different ocean basins, with light shaded symbols representing UVIC_UW3 and dark shaded ones UVIC_FCT. For each symbol plotted, the azimuthal angle is the correlation coefficient between the simulated and observed field, with a correlation coefficient of "1" lying on the $x$-axis, and the radial distance from the origin is the spatial standard deviation of that field. To avoid cluttering the diagram we plot the standard deviation of the corresponding observation as a colored, dashed line so that the observation lies on the intersection of that line with the $x$-axis. The distance between that intersection point and any plotted symbol is the centered RMS difference between the model and data. For clarity these RMS values are also tabulated in the figure.





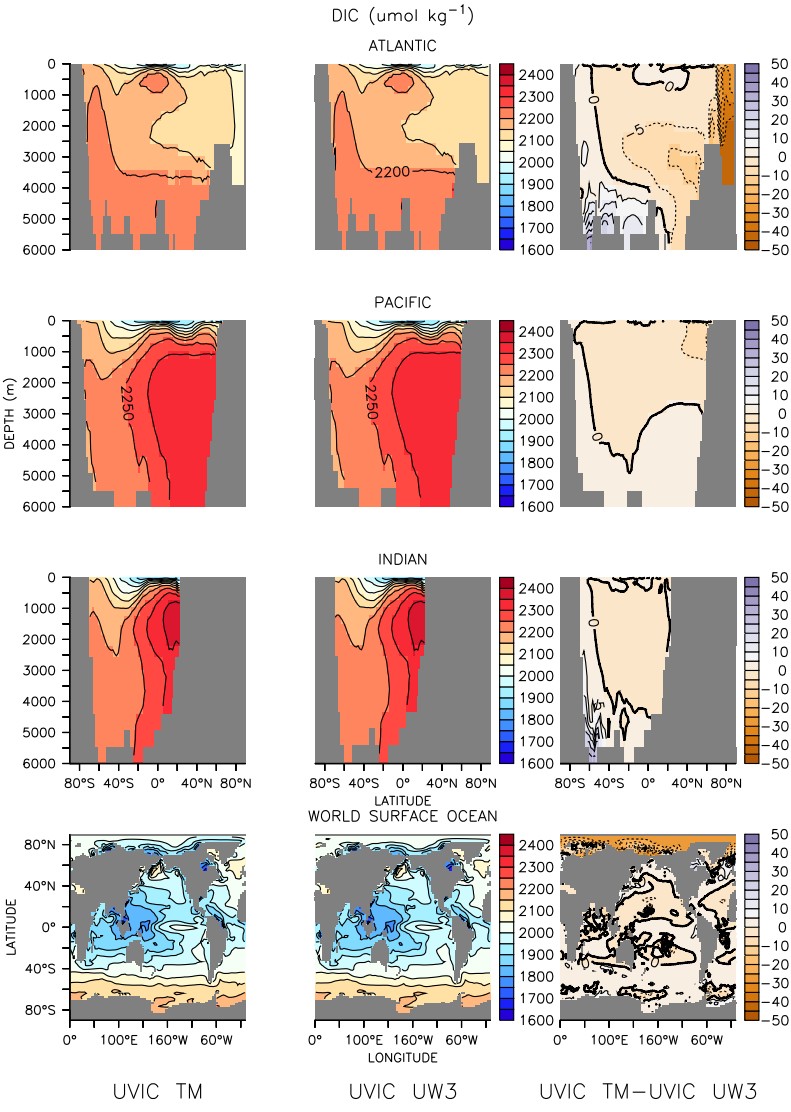

**Figure 6.** Dissolved inorganic carbon in the offline simulation (left column), online simulation (center) and difference between the two (right).





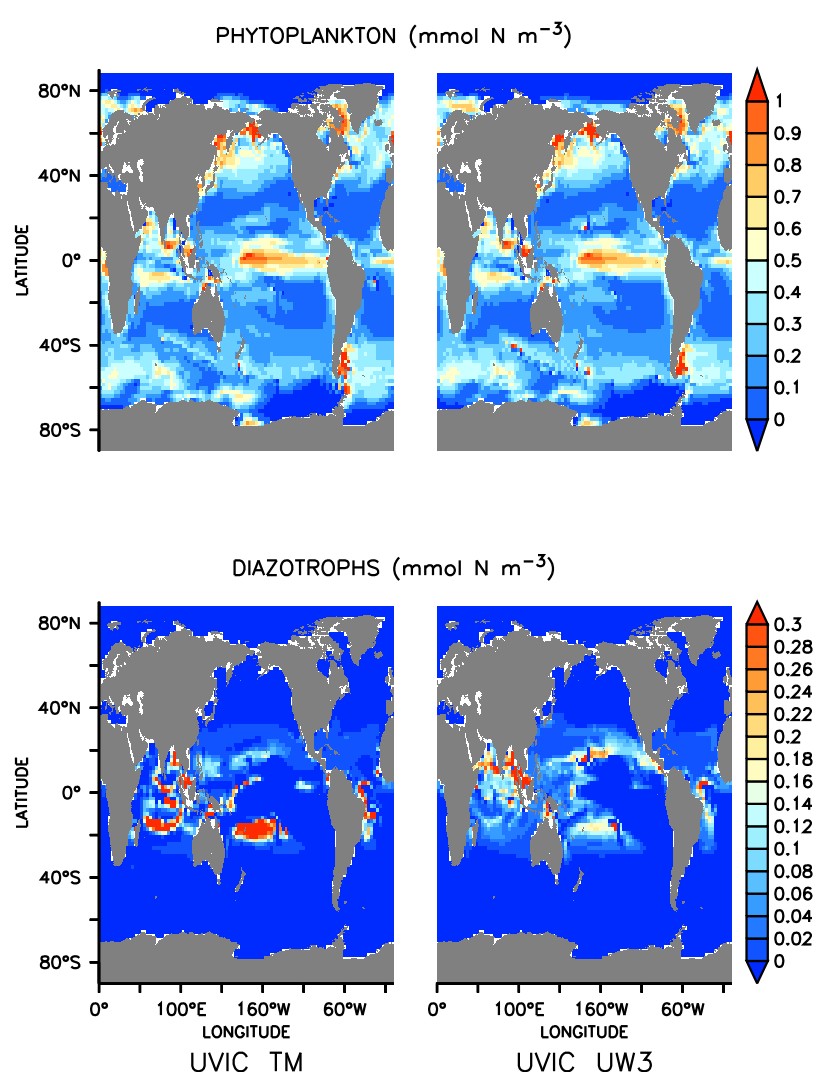

**Figure 7.** Annually averaged surface phytoplankton (top) and diazotroph (bottom) biomass in mmol N m$^{-3}$.





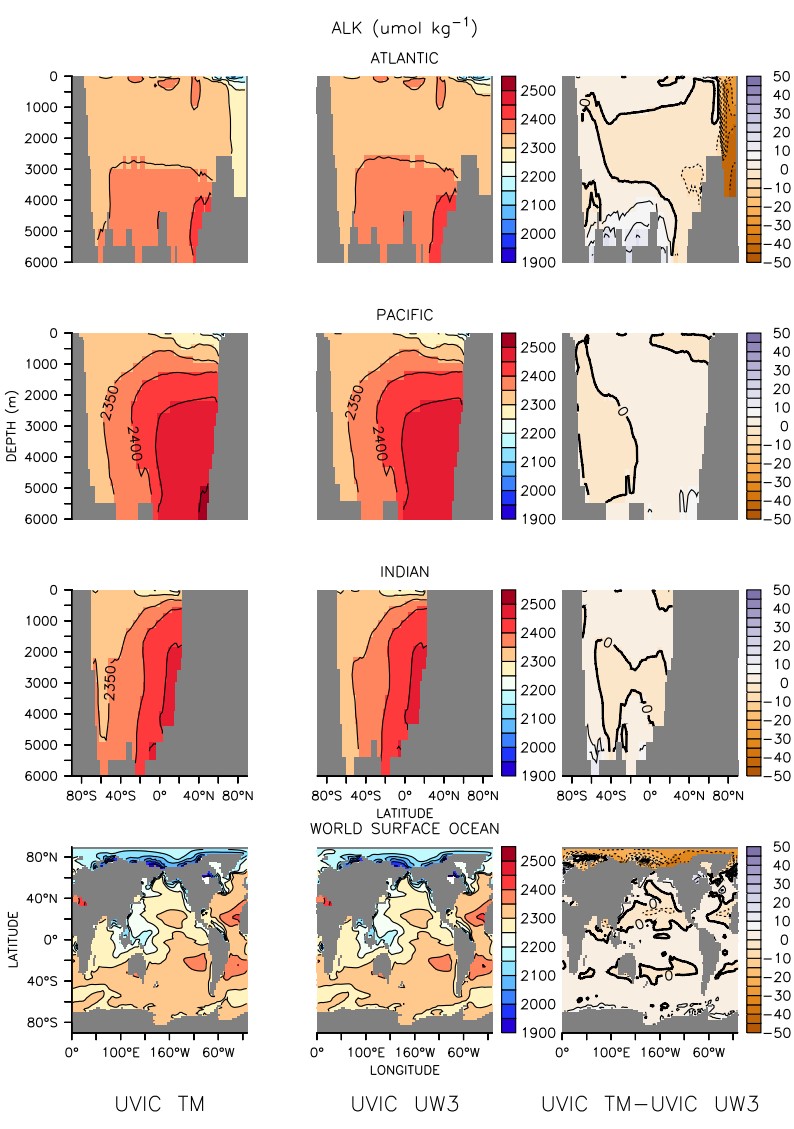

**Figure 8.** Same as Fig. 6 but for alkalinity.





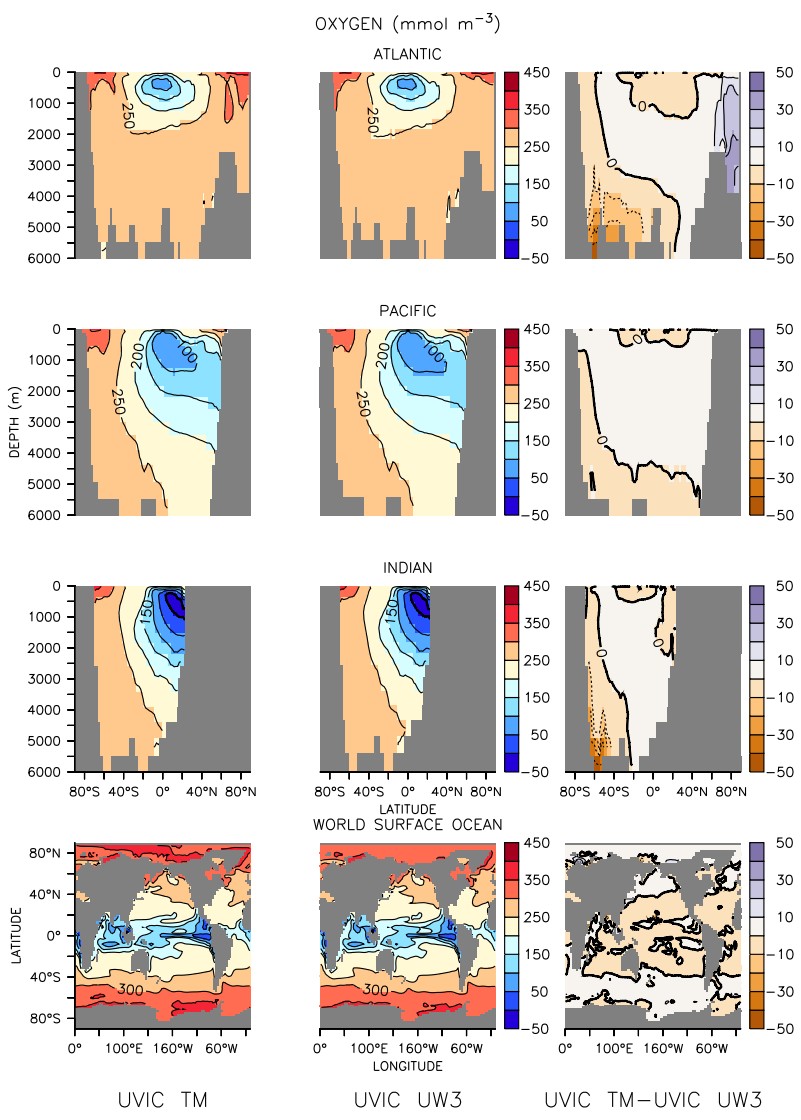

**Figure 9.** Same as Fig. 6 but for oxygen.





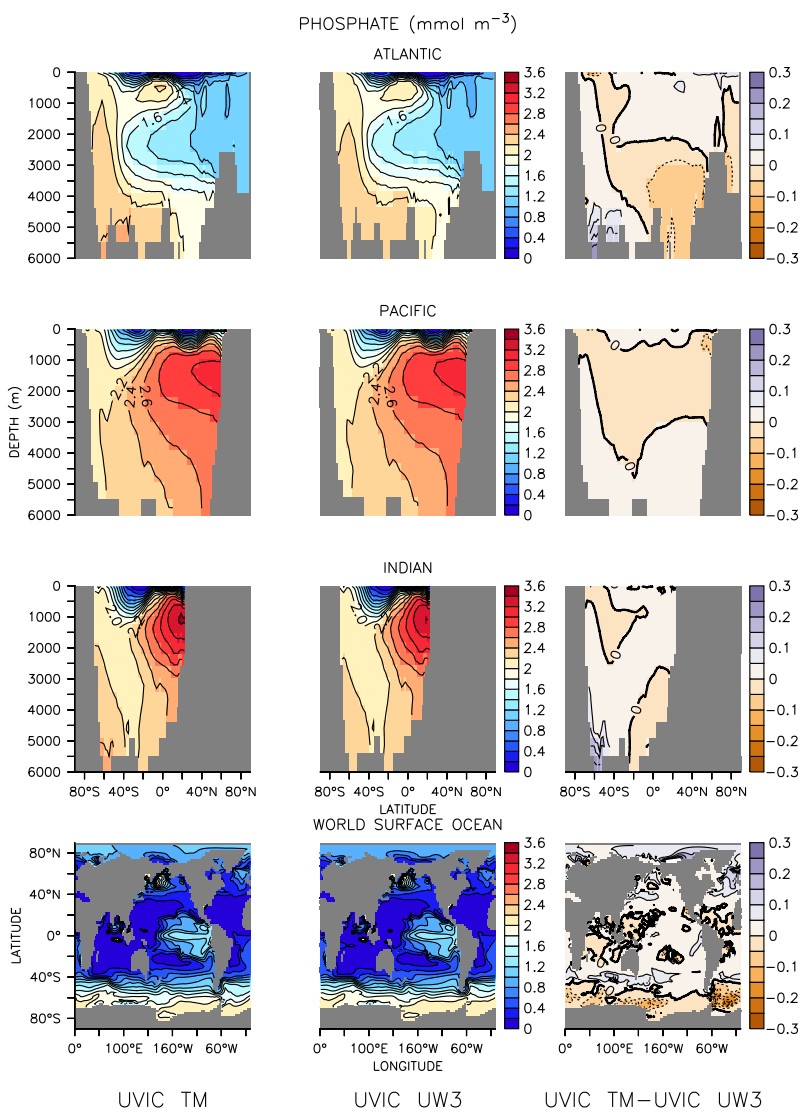

**Figure 10.** Same as Fig. 6 but for phosphate.





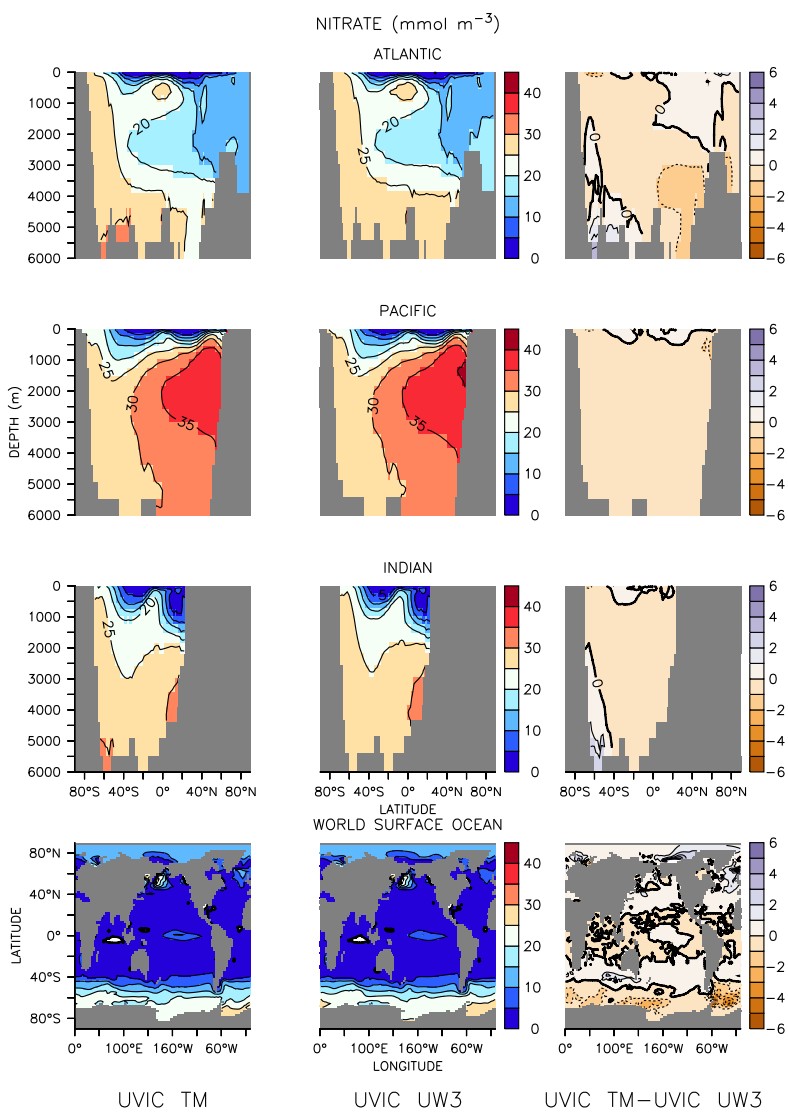

**Figure 11.** Same as Fig. 6 but for nitrate.





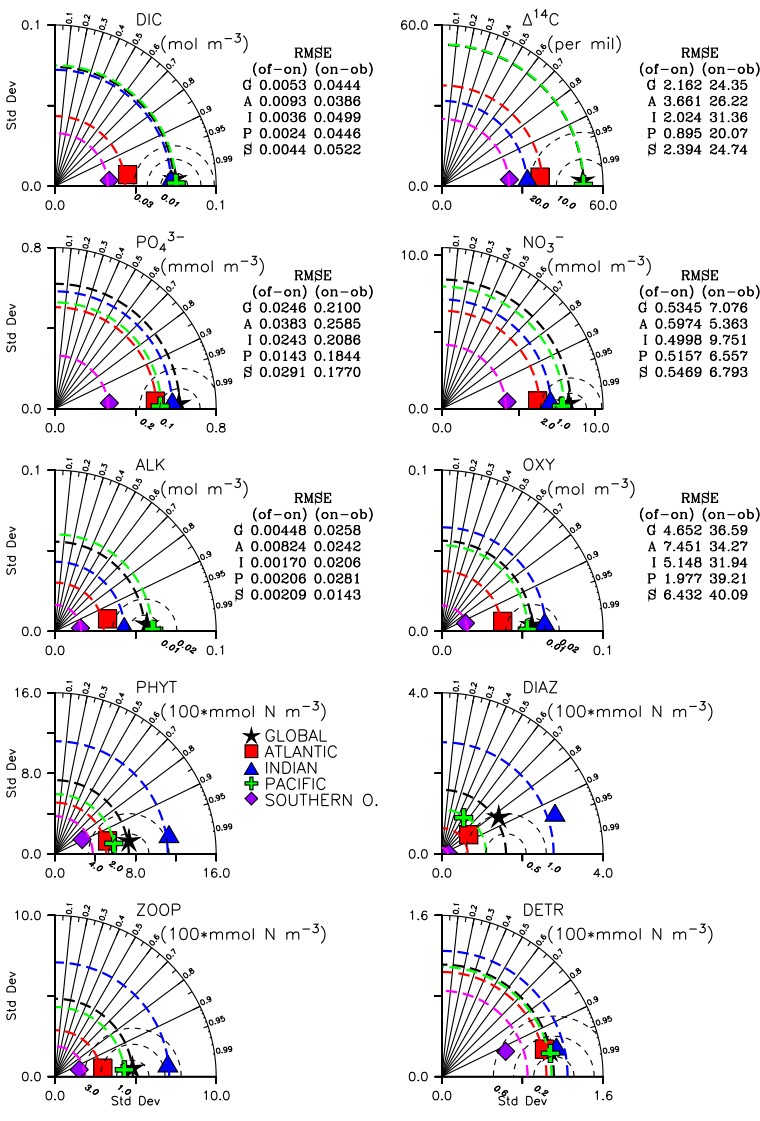

**Figure 12.** Taylor diagrams for various simulated biogeochemical quantities in the offline model compared with the online one. Different symbols and colors correspond to different ocean basins. For each symbol plotted, the azimuthal angle is the correlation coefficient between the offline and online field, with a correlation coefficient of "1" lying on the $x$-axis, and the radial distance from the origin is the spatial standard deviation of the field in the offline model. To avoid cluttering the diagram we plot the standard deviation of the online field as a colored, dashed line so that the online model is the intersection of that line with the $x$-axis. The distance between that intersection point and any plotted symbol is the centered RMS difference between the offline and online models. The RMS differences are also shown in a table next to each plot along with, for context, the centered RMS difference between the online model and observations (where available).





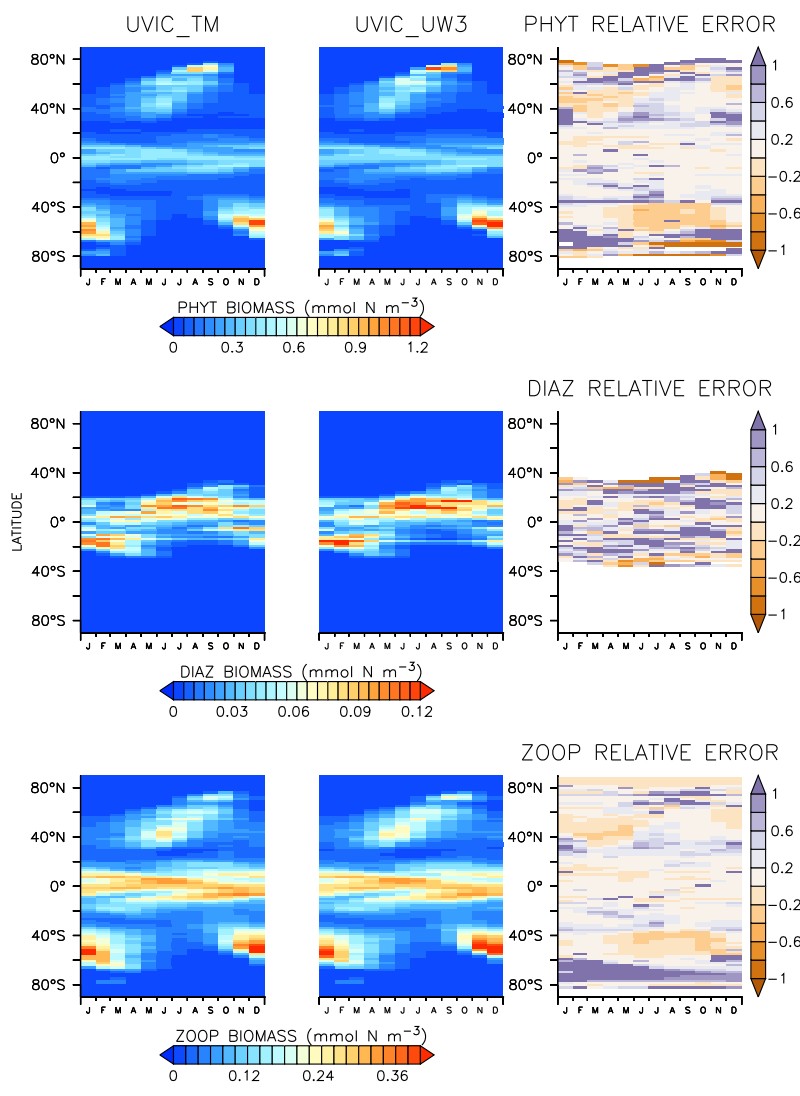

**Figure 13.** Zonally-averaged surface biomass for phytoplankton (top), diazotrophs (middle) and zooplankton (bottom) as a function of month and latitude in the offline simulation (left column), online simulation (center) and relative difference between the two (right). Relative error is calculated as (UVIC_TM - UVIC_UW3)/UVIC_UW3. Units are mmol N m$^{-3}$.





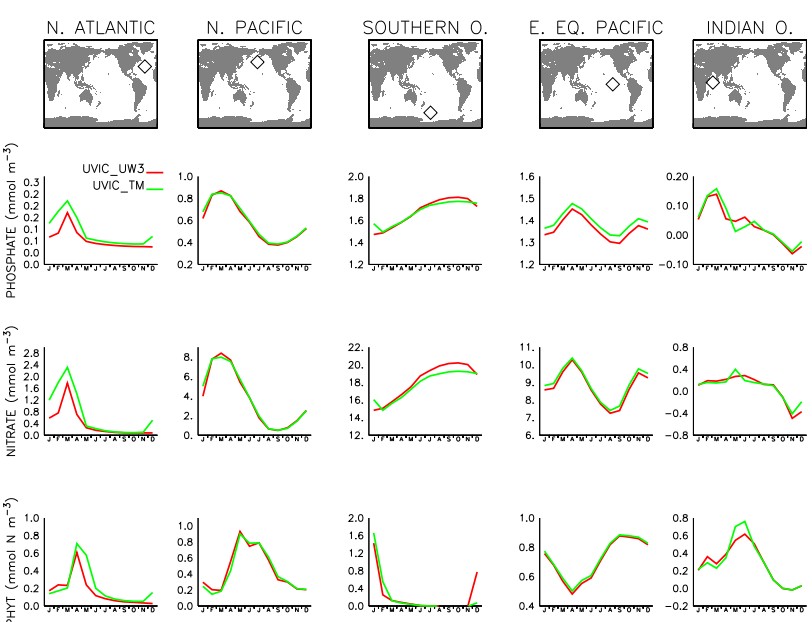

**Figure 14.** Monthly-mean surface phosphate, nitrate, and phytoplankton biomass as a function of calendar month for selected points. The map view indicates the location of the points as diamonds. Models are shown with colors: red is UVIC_UW3, green is UVIC_TM.