# Peer review of "Evaluation of the Transport Matrix Method for simulation of ocean biogeochemical tracers"

_Geoscientific Model Development, 2017_

## Referee Comment (RC1) · Anonymous Referee #1 · 6 Mar 2017

General Comments:

This paper provides relevant scientific information on the differences in biogeochemical tracers between running a biogeochemical model in conjunction with an online OGCM versus running the same biogeochemical model offline using the Transport Matrix Method (TTM), when the transport matrices are based on exactly the same OGCM. Comparisons are made for key biogeochemical tracers (nitrate, phosphate, oxygen) and for phytoplankton biomass (diazothophs, other phytoplankton). The authors conclude that the differences are relatively minor compared to the differences between observations and the modeled values.

For the OGCM used, the University of Victoria Earth System Climate Model, an additional benefit of using the TTM approach is the feasibility of parallelism in the implementation. This resulted in two orders of magnitude gain in the wall-clock time needed to run the biogeochemical model.

The paper is well organized, and quite readable. The use of Taylor diagrams to contrast the results is helpful in summarizing the differences. There is sufficient detail about how this was implemented to facilitate reproducing the results.

It is unfortunate that basic changes to the advection scheme of the online OGCM and a compensating parameter change in a diffusion parameter were required in order to facilitate generation of the transport matrices. That means that the transport matrices are derived from a different version of the OGCM than the version with which the biogeochemical model is usually run. Comparisons of the results using the two OGCM versions are given; and comparisons of results using the modified OGCM and the TTM model are given. What is missing, as a practical matter, is a comparison of the biogeochemical results using the standard OGCM with the results using the TTM model.

Specific Comments:

Page 1, Line 9, forward: The paper abstract needs to inform the reader that modifications to the OGCM were necessary in order to make it feasible to extract the TMs.

Pg. 2, line 22, forward: This line implies that the ocean component of the ESCM was always run along with the atmosphere-biosphere-cryosphere-geosphere, since there is no statement to the contrary. Is this correct? If not, state which components were run in conjunction with the ocean model. This is particularly important in order to set the context for the great gain in the computer time that was made using the TTM.

Were the forcings being used representative of the current era, without increased warming? Please state what forcing scenario was used.

Please give a little more in the description of the biogeochemical model; such as NZPD (declared later), and what phytoplankton groups, grazers, and nutrients are being tracked.

Page 6, oxygen: It is mentioned that diazotrophs are disproportionately sensitive to low oxygen levels, as denitrification can be triggered. Small differences spatially in suboxic conditions can have significant impact. The extent of these differences between the OGCM and TTM in the modeled low oxygen, and where the low oxygen regions occur, needs to be shown.

Technical Corrections:

Pg. 1, Line 8: Insert "course-resolution" after "widely used".

Pg. 1, Line 9: Replace "for" with "from".

Pg. 1, Line 23: Replace "GCM" with "OGCM".

Pg. 2, Line 24: Delete "bug fixes and".

Pg. 4, Line 4: If possible, explain the effect of this (such as making the TMM model 5 times slower), so that the trade-off can be better understood.

Pg. 5, Line 21: Modify "the impact of the advection scheme" to "the impact of the differences in the advection schemes" (or similar).

Pg. 6, Line 33: Insert before "offline and online models", " the biogeochemical component of the".

Pg. 7, Line 1: Insert after "monthly in these experiments", "compared to [what time period] in the online simulation".

Pg. 7, Line 16: "it's" should be "its".

Pg. 7, Line 24: End the sentence "with respect to observations". Period. The hundred-fold improvement in the time is a separate benefit of the particular implementation because of the added parallelization.

Pg. 7, Line 28: Insert "biogeochemistry" at the beginning of the line, before " in a few hours".

Pg. 7, Lines 19 forward: It might enhance the clarity of the conclusions to split this into two paragraphs, one on the biogeochemical results comparison, and another on the large improvement in the computer time required.

Pg. 13, Fig. 2, forward: State that the top panels are zonal averages (assuming that they are).

———————————————

---

## Referee Comment (RC2) · Anonymous Referee #2 · 15 Mar 2017

A comparison of on-line and offline models, the latter using the Transport Matrix Method, is long overdue and will be a welcome addition to the literature, even just as an example for a single model. It's an easy paper to review as the requirements for such a paper are just 2 things: a good description of the method; a sensible choice of parameters to compare. I'm keen to see the paper published but there are a few details that I'd like to see addressed before then...

Method:

- I don't have an issue with the choice of model as the paper is effectively an example and I'm not sure I see the value in a much longer and exhaustive paper doing the same with a variety of models, particularly as they are all evolving. However, I don't see the value of the comparison of how the model performs with and without the FCT scheme.

[Figure]

For the purposes of the paper all that is needed is a base model – it doesn't matter if it performs a little less well than another version. Hence, I would either cut Section 3.1 or move it into an Appendix. As another option, if it is argued that Section 3.1 is there to allow comparison of the offline run to the FCT case then this needs to be done more rigorously by taking the spun-up UW3 model and running it onwards on-line with FCT now turned on for the comparison.

- The manuscript is a little vague about the details of the runs for the comparison of the on-line models. The starting point is a 13,000 year spin-up of the on-line UW3 model. The TM is then extracted using an extra year run. Are the offline and on-line models then compared purely on the basis of a single extra year run after the 13,000 year spin-up? I would hope not as unless the off-line perfectly mimics the on-line model there is no guarantee that any transient response of the off-line model will be fast, and the differences may be small purely because the two models have had little time to diverge, particularly if it takes 13,000 years to spin-up. The comparison of the difference between the on-line and off-line models should at least state the time over which the models are run for comparison and this should be at least of order 10 years. My recommendation for minor changes is on the assumption that the models were run for longer than 1 year before comparison. If just one year I think the need to re-run the models for longer would constitute major changes

- The manuscript describes the issue with a leapfrog scheme but is a little vague about the compromise made. Is the TM extracted from just one strand of the leapfrog scheme? Is the usual process of blending the 2 strands side-stepped and ignored for the TM?

Comparison

- The authors suggest that some of the largest fractional errors come from differences in small values. In Fig. 7 though it looks like there are significant errors associated with large values for diazotrophs. I'd like to see an extra column of plots showing the

difference between the on-line and off-line models for these fields as elsewhere in the manuscript.

- The relative errors in Fig 13 are very noisy. It would help to additionally have the contours for +/-0.5 relative error marked on the panels for on-line and off-line zonally averaged fields.

- A couple of minor points but the y-axis labelling on Fig 14 needs an extra decimal place for phosphate and it also seems strange that phosphate diverges (then converges) so quickly from the same starting point if it is just a one year run.

---

## Referee Comment (RC3) · Anonymous Referee #3 · 22 Mar 2017

In this study, the authors extract a transport matrix model (TMM) from the ocean component of the UVic Earth System Climate model, and use the TMM to spin-up the biogeochemical component of the model. The biogeochemical state of the TMM-model is then compared to an identical simulation using the online circulation model. Overall, this paper provides a useful comparison of the TMM spin-up method to the online method, and shows that the TMM faithfully represents most aspects of the online model, at a fraction of the computational time.

This paper is well-written and appropriate for publication in Geoscientific Model Development. However, there are quite a few points in the paper that need to be expanded on and/or clarified in a revised manuscript. These are listed below.

Page 4, line 2: "time-stepped with a simple Euler method": Euler forward or Euler

backward?

Page 4, line 3: "Fourier filtering at high latitudes": Some more discussion of this would be useful. Why is this filtering applied? What is the underlying cause of the noisiness? What numerical grid scheme (e.g. Arakawa B, C etc.) does the model use?

Page 4, line 7: "Monthly mean TMMs were extracted": More information about how this was done is needed here. What time-step was used in the online model to create the TMs? How was the monthly averaging done?

Page 4, line 20 ff.: These few lines of description are not sufficient. More information is needed here to better describe how the biogeochemical model is coupled to the TMM. Equations and/or pseudocode would be appropriate so that one does not have to download and wade through the code.

Page 4, line 30: Why does the MOC weekend when switching from the FCT to UW3 advection schemes? Some discussion of this is needed.

Page 5, line 25: Why is alkalinity sensitive to small changes in oxygen?

Page 5, "Mean state" section: Define the "mean state" of the TMM model and the online model. Are they directly comparable? Presumably the "mean state" of the TMM model is the annual average of the seasonally-cycling model which represents year 13001 i the online model — is that correct? Is the mean state of the online annual average of the 13001st model year? Or is it the multi-annual average of some range of years — and thus would include natural inter annual variability as well? This needs a careful description, and if the two "mean states" are not directly comparable, this should be discussed.

Figure 9: Oxygen: It would be useful also to show the suboxic/hypoxic volume for the TMM and online models. Do they match up well? And related to this, the water-column denitrification rate in each model — how does it compare? This is an important biogeochemical process that is highly sensitive to the details of the oxygen distribution.

It is important to know if the TMM version of the model captures the behavior of the online model.

Page 6, line 15, and Figures 10 and 11: "Polar filtering" is blamed for the mismatch at high latitudes. Seems likely that this is not the cause. The mismatch is not really in the "polar" regions - is this filtering really applied at 50-60oS in the ACC? Also, the nutrients are too high in the surface S. Ocean and too low in the deep S. Ocean — this seems to implicate the biological pump (e.g. particle formation/sinking) as the culprit. A more careful discussion of these differences and their possible causes is warranted.

Section 3.2.2 "Seasonal cycle". More discussion of how the seasonal cycle is handled is needed. Equations are needed. Is there a separate TM for each month? And then Euler forward (or backward) is applied to time-step the model? I'm assuming this is the case, but this should be made explicit. In regards to the difference between the TMM and online model, for example as seen in the Indian Ocean for phosphate (Fig. 14), how much is due to the time-averaging of the TMM, and how much to the time-stepping scheme? Would some of these differences be reduced with a more robust time-stepping scheme? e.g. Adams-Bashforth or Crank-Nicholson.

---

## Author Comment (AC1) · 3 May 2017

Author response to comments of Reviewer 1:

We would like to thank the reviewer for their efforts in helping us to improve our manuscript. Referee remarks are shown in red and our responses are given in black font. Changes to the manuscript text are given in blue font.

General Comments:
This paper provides relevant scientific information on the differences in biogeochemical tracers between running a biogeochemical model in conjunction with an online OGCM versus running the same biogeochemical model offline using the Transport Matrix Method (TTM), when the transport matrices are based on exactly the same OGCM. Comparisons are made for key biogeochemical tracers (nitrate, phosphate, oxygen) and for phytoplankton biomass (diazothophs, other phytoplankton). The authors conclude that the differences are relatively minor compared to the differences between observations and the modeled values.

For the OGCM used, the University of Victoria Earth System Climate Model, an additional benefit of using the TTM approach is the feasibility of parallelism in the implementation. This resulted in two orders of magnitude gain in the wall-clock time needed to run the biogeochemical model.

The paper is well organized, and quite readable. The use of Taylor diagrams to contrast the results is helpful in summarizing the differences. There is sufficient detail about how this was implemented to facilitate reproducing the results.

It is unfortunate that basic changes to the advection scheme of the online OGCM and a compensating parameter change in a diffusion parameter were required in order to facilitate generation of the transport matrices. That means that the transport matrices are derived from a different version of the OGCM than the version with which the biogeochemical model is usually run. Comparisons of the results using the two OGCM versions are given; and comparisons of results using the modified OGCM and the TTM model are given. What is missing, as a practical matter, is a comparison of the biogeochemical results using the standard OGCM with the results using the TTM model.

We would not regard changing the scheme to suit our purpose "unfortunate". (More of a "nuisance".) There is no single, perfect advection scheme (or, more generally, numerical method). And while it is true that historically UVic ESCM has been run with a particular nonlinear advection scheme (which the reviewer calls "standard"), that is a choice dictated by a variety of factors and the model parameters had to be tuned appropriately. There is no reason to not use a different scheme if circumstances call for it. Practical choices like this are made all the time in scientific computing.

Regarding the reviewer's suggestion to include a comparison of biogeochemical results from the "standard" OGCM with the TMM, we emphasize that the primary purpose of our manuscript is to provide a comparison between identical online and offline biogeochemical model runs in order to assess the offline method. This is why we did not spend much time re-tuning the physical model beyond increasing the vertical diffusion parameter to stabilise the overturning circulation. The reason we included a comparison between online runs of the OGCM performed with the two different advection schemes is that we recognize that other users of the UVic ESCM who may be interested in using the offline method, but currently use the default nonlinear advection scheme, will want to know about the consequences of switching advection schemes. But, again, it is not the main purpose of the paper. Moreover, we also note that one of the other referees thought that it wasn't even necessary to

present this comparison, and that it be either removed or, at the very least, moved to an appendix. We agree that it does distract from the main goal of our paper and have followed that reviewer's advice and moved it to an appendix.

Specific Comments:

Page 1, Line 9, forward: The paper abstract needs to inform the reader that modifications to the OGCM were necessary in order to make it feasible to extract the TMs.

The following sentence has been added to Line 9:

The default, non-linear advection scheme was first replaced with a linear, third order upwind-biased advection scheme to satisfy the linearity requirement of the TMM.

Pg. 2, line 22, forward: This line implies that the ocean component of the ESCM was always run along with the atmosphere-biosphere-cryosphere-geosphere, since there is no statement to the contrary. Is this correct? If not, state which components were run in conjunction with the ocean model. This is particularly important in order to set the context for the great gain in the computer time that was made using the TTM.

In addition to the ocean GCM and biogeochemistry, the UVic ESCM has a prognostic sea ice model, atmospheric energy balance model (EBM) and land biosphere component. All of these were switched on for the online runs. Switching off the land model make very little difference to the computational time (although in our prescribed $CO_2$-experiments it was not necessary to turn it on, an oversight on our part). The EBM takes up about 20% of the computational time but, as currently implemented, it is not possible to switch it off and drive the OGCM with prescribed surface fluxes. (Of course, with appropriate code modifications it should be possible to do this.) To clarify, Sec. 3.2 has been modified to read:

The UVic ESCM is a serial code and thus unable to exploit more than one computational core. With the biogeochemical component switched on the model throughput on a typical Linux machine is about ~250 model years per day. (It should be noted that the model was run with the atmospheric energy balance model (EBM) switched on. This adds roughly 20% to the computational cost of running the model. However, as currently implemented, it is not possible to switch off the EBM in UVic ESCM and drive the ocean GCM with prescribed fluxes.) A 5000-year spin-up of the online biogeochemical model thus takes ~3 weeks. On the other hand, the PETSc-based TMM version can run in parallel, even though the underlying biogeochemical code is identical. While we have not carried out a detailed scaling analysis of the TMM version's performance, a similar 5000-year spin-up can be accomplished in 3.8 hours with 256 cores on NCAR's Yellowstone IBM iDataPlex cluster, and in 5.2 hours with 160 cores on GEOMAR/Kiel University's NEC HPC machine.

Were the forcings being used representative of the current era, without increased warming? Please state what forcing scenario was used.

As stated on Page 4, line 34, all simulations were carried out with a fixed, pre-industrial atmospheric $CO_2$ concentration of 277.4 ppm. The forcings are thus representative of that period.

Please give a little more in the description of the biogeochemical model; such as NZPD (declared later), and what phytoplankton groups, grazers, and nutrients are being tracked.

Page 2, line 24 has been amended and expanded to read:

The marine nutrients-phytoplankton-zooplankton-detritus (NPZD) biogeochemistry has increased in complexity since Eby et al. (2009), with the addition of iron limitation and revisions to zooplankton grazing (Keller et al., 2012), and subsequent minor updates. The NPZD model contains two phytoplankton types (a general type and diazotrophs) and a single zooplankton type, DIC, alkalinity, nitrate, phosphate (the base unit), and oxygen. Iron limitation is prescribed using a seasonally varying iron mask. Full model details can be found in Keller et al. (2012) and associated references.

Page 6, oxygen: It is mentioned that diazotrophs are disproportionately sensitive to low oxygen levels, as denitrification can be triggered. Small differences spatially in suboxic conditions can have significant impact. The extent of these differences between the OGCM and TTM in the modeled low oxygen, and where the low oxygen regions occur, needs to be shown.

Figure 9 (now Figure 4) has been changed to include suboxic regions (less than 5 mmol m3), and the depth of the map plots is lowered to 300 m.

Technical Corrections:
Pg. 1, Line 8: Insert "course-resolution" after "widely used".

Done

Pg. 1, Line 9: Replace "for" with "from".

Done

Pg. 1, Line 23: Replace "GCM" with "OGCM".

Done

Pg. 2, Line 24: Delete "bug fixes and".

Done

Pg. 4, Line 4: If possible, explain the effect of this (such as making the TMM model 5 times slower), so that the trade-off can be better understood.

We have modified the relevant paragraph in Sec. 2.1, which now reads as:

Lastly, UVic ESCM applies Fourier filtering in the zonal direction at high latitudes to remove grid-scale noise. The efficiency of the TMM arises from the fact that the discretized advection-diffusion operator has a limited stencil, i.e., only couples nearby points, giving rise to a sparse matrix. Fourier filtering on the other hand couples all points in the zonal direction, greatly reducing the sparsity of the transport matrix and hence the computational efficiency of the sparse matrix-vector products at the heart of the TMM. While the cost of a sparse matrix-vector product is implementation- and hardware-dependent and non-trivial to analyze (e.g., Gropp et al., 2000), it roughly scales with the number of non-zero elements per row. With a $3^{rd}$ order upwind scheme, there are a maximum of 5 x 5 x 5 = 125 non-zero elements per row. With Fourier filtering that becomes nx x 5 x 5, where nx is the number of zonal grid points. In UVic ESCM, nx=100, implying that the TMM would be roughly

nx/5=20 times slower with Fourier filtering turned on. We therefore turn off polar filtering for the passive tracers used to extract the TMs. The numerical treatment of temperature and salinity by the model is not altered.

Pg. 5, Line 21: Modify "the impact of the advection scheme" to "the impact of the differences in the advection schemes" (or similar).

Changed to:

"highlight the impact of different advection schemes"

Pg. 6, Line 33: Insert before "offline and online models", " the biogeochemical component of the".
Done

Pg. 7, Line 1: Insert after "monthly in these experiments", "compared to [what time period] in the online simulation".

We're not sure what the reviewer is getting at here. There is no averaging necessary in the online model.

Pg. 7, Line 16: "it's" should be "its".

Done

Pg. 7, Line 24: End the sentence "with respect to observations". Period. The hundredfold improvement in the time is a separate benefit of the particular implementation because of the added parallelization.

Done

Pg. 7, Line 28: Insert "biogeochemistry" at the beginning of the line, before " in a few hours".

Done

Pg. 7, Lines 19 forward: It might enhance the clarity of the conclusions to split this into two paragraphs, one on the biogeochemical results comparison, and another on the large improvement in the computer time required.

Done

Pg. 13, Fig. 2, forward: State that the top panels are zonal averages (assuming that they are).

Done

---

## Author Comment (AC2) · 3 May 2017

Author response to comments from Referee 2:

We thank the reviewer for their thoughtful review. Referee remarks are shown in red and our responses are given in black font. Changes to the manuscript text are given in blue font.

A comparison of on-line and offline models, the latter using the Transport Matrix Method, is long overdue and will be a welcome addition to the literature, even just as an example for a single model. It's an easy paper to review as the requirements for such a paper are just 2 things: a good description of the method; a sensible choice of parameters to compare. I'm keen to see the paper published but there are a few details that I'd like to see addressed before then...

Method:

- I don't have an issue with the choice of model as the paper is effectively an example and I'm not sure I see the value in a much longer and exhaustive paper doing the same with a variety of models, particularly as they are all evolving. However, I don't see the value of the comparison of how the model performs with and without the FCT scheme.

For the purposes of the paper all that is needed is a base model – it doesn't matter if it performs a little less well than another version. Hence, I would either cut Section 3.1 or move it into an Appendix. As another option, if it is argued that Section 3.1 is there to allow comparison of the offline run to the FCT case then this needs to be done more rigorously by taking the spun-up UW3 model and running it onwards on-line with FCT now turned on for the comparison.

We agree with the reviewer that this comparison detracts from the main purpose of the paper and only included it because it could be of interest to other UVic users who may want to apply the TMM. We have moved Section 3.1 to an Appendix.

- The manuscript is a little vague about the details of the runs for the comparison of the on-line models. The starting point is a 13,000 year spin-up of the on-line UW3 model. The TM is then extracted using an extra year run. Are the offline and on-line models then compared purely on the basis of a single extra year run after the 13,000 year spin-up? I would hope not as unless the off-line perfectly mimics the on-line model there is no guarantee that any transient response of the off-line model will be fast, and the differences may be small purely because the two models have had little time to diverge, particularly if it takes 13,000 years to spin-up. The comparison of the difference between the on-line and off-line models should at least state the time over which the models are run for comparison and this should be at least of order 10 years. My recommendation for minor changes is on the assumption that the models were run for longer than 1 year before comparison. If just one year I think the need to re-run the models for longer would constitute major changes

We're sorry for the confusion. The TMs are extracted from an equilibrium state of the OGCM (obtained after a 13,000 year integration). This only requires running the OGCM for one additional year. The TMs are subsequently used to perform an offline integration of the biogeochemical model for 5000 years to equilibrium. Output from the final year of this run is used in the online/offline comparison. We have added the

following paragraph to the end of Section 2.2:

The offline biogeochemical model is forced with the relevant physical and biogeochemical fields taken from the equilibrated online model. In the present case, these are monthly mean wind speed, insolation, sea ice concentration, temperature, salinity, freshwater flux (evaporation, precipitation and runoff) and iron concentration. All fields, including the previously extracted transport matrices (also at monthly mean resolution), are linearly interpolated to the current time step before being applied. The offline model was integrated with a time step of 8 hours for 5000 years to equilibrium, with monthly averages of various fields from the final year of this run used for comparison with the equilibrated online simulation.

- The manuscript describes the issue with a leapfrog scheme but is a little vague about the compromise made. Is the TM extracted from just one strand of the leapfrog scheme? Is the usual process of blending the 2 strands side-stepped and ignored for the TM?

We apologize for being a bit vague on this point. We have modified the relevant paragraph in Sec. 2.1 as follows to provide additional detail:

A second complication is from the time-stepping scheme, which in UVic ESCM is leapfrog. The explicit horizontal advection and diffusion terms are also sometimes staggered with respect to each other for stability. Both require storing the tracer field at odd and even time steps. While this can be replicated offline, in order to use a common scheme for all ocean models from which TMs have been extracted (e.g., MITgcm variously uses Adams-Bashforth, direct space-time discretization and other schemes), we combine horizontal advection and diffusion into a single explicit transport matrix, $A_e$ which is time-stepped with a simple, forward Euler method. Specifically, to extract the explicit matrix, we only store the (passive) tracer field at the current time step. This field, which is reset to a pattern of 1's and 0's at the beginning of each time step, is then stepped forward by UVic ESCM like any other tracer. The change in the tracer field divided by the time step is the explicit tendency matrix. With this procedure, which does not require changes to the underlying code, the usual leapfrog scheme is side-stepped.

Comparison

- The authors suggest that some of the largest fractional errors come from differences in small values. In Fig. 7 though it looks like there are significant errors associated with large values for diazotrophs. I'd like to see an extra column of plots showing the difference between the on-line and off-line models for these fields as elsewhere in the manuscript.

We have modified the figure (now Figure 2) to additionally show offline-online differences.

- The relative errors in Fig 13 are very noisy. It would help to additionally have the contours for +/-0.5 relative error marked on the panels for on-line and off-line zonally averaged fields.

Figure 13 is now Figure 8. Including contours as suggested made the plot just as

noisy, but changing the format from "shade" to "fill" (these plots are made in Ferret) and adjusting the scale has reduced their noise.

- A couple of minor points but the y-axis labelling on Fig 14 needs an extra decimal place for phosphate and it also seems strange that phosphate diverges (then converges) so quickly from the same starting point if it is just a one year run.

Figure 14 is now Figure 9. The y axis of phosphate has been fixed. Slight differences in the application of external forcing between online and offline models are the likely cause of differences between the simulations, particularly in regions experiencing high seasonality (e.g., the North Atlantic).

We're don't follow the reviewer's comment that "it is just a one year run". Each line on the plot represents an equilibrium solution obtained after a long spin-up integration (13,000 years for the online run and 5000 years for the offline run).

---

## Author Comment (AC3) · 3 May 2017

Author response to comments from Referee 3:

We thank the referee for their insightful review. Referee remarks are shown in red and our responses are given in black font. Changes to the manuscript text are given in blue font.

In this study, the authors extract a transport matrix model (TMM) from the ocean component of the UVic Earth System Climate model, and use the TMM to spin-up the biogeochemical component of the model. The biogeochemical state of the TMM model is then compared to an identical simulation using the online circulation model. Overall, this paper provides a useful comparison of the TMM spin-up method to the online method, and shows that the TMM faithfully represents most aspects of the online model, at a fraction of the computational time.

This paper is well written and appropriate for publication in Geoscientific Model Development. However, there are quite a few points in the paper that need to be expanded on and/or clarified in a revised manuscript. These are listed below.

Page 4, line 2: "time-stepped with a simple Euler method": Euler forward or Euler backward?

We use Euler forward. The text has been changed accordingly.

Page 4, line 3: "Fourier filtering at high latitudes": Some more discussion of this would be useful. Why is this filtering applied? What is the underlying cause of the noisiness? What numerical grid scheme (e.g. Arakawa B, C etc.) does the model use?

UVic's ocean model is based on MOM2 and uses an Arakawa B grid. Reviewer 1 also raised the issue of Fourier filtering. We have modified the relevant paragraph in Sec. 2.1 to the following:

Lastly, UVic ESCM applies Fourier filtering in the zonal direction at high latitudes to remove grid-scale noise. The efficiency of the TMM arises from the fact that the discretized advection-diffusion operator has a limited stencil, i.e., only couples nearby points, giving rise to a sparse matrix. Fourier filtering on the other hand couples all points in the zonal direction, greatly reducing the sparsity of the transport matrix and hence the computational efficiency of the sparse matrix-vector products at the heart of the TMM. While the cost of a sparse matrix-vector product is implementation- and hardware-dependent and non-trivial to analyze (e.g., Gropp et al., 2000), it roughly scales with the number of non-zero elements per row. With a 3$^{rd}$ order upwind scheme, there are a maximum of 5 x 5 x 5 = 125 non-zero elements per row. With Fourier filtering that becomes nx x 5 x 5, where nx is the number of zonal grid points. In UVic ESCM, nx=100, implying that the TMM would be roughly nx/5=20 times slower with Fourier filtering turned on. We therefore turn off polar filtering for the passive tracers used to extract the TMs. The numerical treatment of temperature and salinity by the model is not altered.

Page 4, line 7: "Monthly mean TMMs were extracted": More information about how this was done is needed here. What time-step was used in the online model to create the TMs? How was the monthly averaging done?

We thank the reviewer for raising this point, which we failed to address in the original manuscript. We have added the following paragraph to Sec. 2.1:

The neglect of polar filtering and staggering of advection and diffusion terms necessitates using, for stability, a slightly smaller time step in offline simulations with the TMM compared with the online model. In the latter, the default time step is 1.25 days, a choice dictated by the need to synchronize the ocean and atmospheric models. (The biogeochemical terms in UVic ESCM are time-stepped internally within the biogeochemical module using a much smaller time step such that there are 3 biogeochemical steps per ocean step.) No change was made to this during extraction of the TMs, i.e., the model physics and active tracers were integrated using the default time step. Since the explicit TM is extracted as a tendency, the time step for offline explicit advection-diffusion can be subsequently set to any desired value. However, the implicit TM has a time step embedded within it. By default it would also be 1.25 days, but embedding a different time step is quite straightforward: during extraction of the implicit TM, we simply pass the desired time step as an argument to the subroutine that solves for implicit diffusion. We have found an offline time step of 8 hours (28,800 s) to be a good compromise between stability and accuracy. It is also very similar to the biogeochemical time step of the online model (27,000 s).

To clarify the issue of time averaging, the last paragraph of Sec. 2.1 has been changed to:

Using the linear UW3 advection scheme, the coupled physical-biogeochemical model was spun-up to equilibrium for 13,000 years with a fixed, pre-industrial atmospheric $CO_2$ concentration of 277.4 ppm. The model was run for one additional year with the transport matrix extraction switched on. During this run, explicit and implicit TMs were computed at each time step, and accumulated over the course of a month before being averaged and written out. These monthly mean TMs were subsequently used for the offline simulations. For comparison, we also carried out a similar spin-up of the physical and biogeochemical model using the default FCT advection scheme (see Appendix A).

Page 4, line 20 ff.: These few lines of description are not sufficient. More information is needed here to better describe how the biogeochemical model is coupled to the TMM. Equations and/or pseudocode would be appropriate so that one does not have to download and wade through the code.

We appreciate the reviewer's point about having to wade through the code to see how we interface the biogeochemical model to the TMM. But we note that there aren't any special "TMM equations" beyond eqn. (1), which shows how the biogeochemical term (q) is incorporated into the TMM framework. In a sense that is all there is to it. Obviously of course there are a lot of implementation details that are very specific to each biogeochemical model. But this makes it difficult to come up with any sort of meaningful "pseudocode" as the reviewer asks for. Instead, we have expanded the discussion in Sec. 2.2 in a way that we hope addresses the reviewer's concerns. It now reads as:

To apply the TMM code to a particular biogeochemical problem essentially requires providing a routine that takes as input vertical ``profiles'' of tracer concentrations at a

horizontal location at the current time step (along with corresponding variables such as layer thickness, temperature, wind speed, etc at that location), and returns profiles of the biogeochemical tendency term, q. In practice, coupling an existing biogeochemical model such as the one in UVic ESCM to the TMM framework involves writing a ``wrapper'' routine that serves as an interface between the TMM driver (written in C) and the biogeochemical code (typically written in some dialect of Fortran).

While the specific implementation of the wrapper will depend on the details of the biogeochemical model, in general it performs three main tasks. First, it copies required data from TMM arrays (that are passed as input arguments to the wrapper routine) to those of the biogeochemical model. Second, it calls the actual routine that computes biogeochemical source/sink terms (q). Normally, this routine would be called from the time-stepping loop of the model in which the biogeochemical model is embedded. Third, as these tendency terms are stored in arrays in the biogeochemical model, the wrapper copies them to arrays that are passed back to the TMM driver (as output arguments to the wrapper routine). To simplify this exchange of data, the horizontal grid on which the biogeochemical model (and ocean model in which it is embedded) is declared to have a size of 1x1. In essence, the biogeochemical model is treated as a 1-dimensional column model. In the case of UVic ESCM, where the code for physical and biogeochemical models are deeply intertwined, a few minor, additional changes to the original code were also necessary. Most of these changes were required in order to make available the full set of diagnostics accumulated by UVic ESCM. See Sec. 5 for information on where to download the code from.

Page 4, line 30: Why does the MOC weekend when switching from the FCT to UW3 advection schemes? Some discussion of this is needed.

It is believed that the strength of the MOC increases with the vertical diffusivity (at least in models). In a numerical model that diffusion arises not only from the explicitly modeled diffusion term (and corresponding prescribed diffusivity), but also from the implicit diffusion inherent to every numerical advection scheme. As is stated in the Methods section on page 3, 3rd paragraph, UW3 is less diffusive than FCT, which may be why the MOC weakens when switching from FCT to UW3. However, while an important and interesting point, addressing it in more detail than already stated in the manuscript would take us beyond the scope of the present study.

Page 5, line 25: Why is alkalinity sensitive to small changes in oxygen?

This section is now an appendix. The sentence has been expanded to read:

The largest differences are in nitrate and alkalinity, both of which are sensitive to small differences in oxygen via biological processes discussed in Section 3.1.

Page 5, "Mean state" section: Define the "mean state" of the TMM model and the online model. Are they directly comparable? Presumably the "mean state" of the TMM model is the annual average of the seasonally-cycling model which represents year 13001 in the online model âA˘T˘ is that correct? Is the mean state of the online annual average of the 13001st model year? Or is it the multi-annual average of some

range of years — and thus would include natural inter annual variability as well? This needs a careful description, and if the two "mean states" are not directly comparable, this should be discussed.

The UVic ESCM has no inter-annual variability and the reviewer is correct that in the online model the mean state is that represented by the final year of the 13,000-year spin-up. In the TMM the mean state is represented by the final year of a 5,000-year spin-up. The first two sentences of Sec. 3.1.1 now read:

We first compare the annual-mean state of the online (UVIC_UW3) and offline (UVIC_TM) simulations, taken from the final year of the corresponding spin-up. A fully stable UVic ESCM simulation with annually repeating seasonal forcing has no inter-annual variability.

Figure 9: Oxygen: It would be useful also to show the suboxic/hypoxic volume for the TMM and online models. Do they match up well? And related to this, the water-column denitrification rate in each model — how does it compare? This is an important biogeochemical process that is highly sensitive to the details of the oxygen distribution.

It is important to know if the TMM version of the model captures the behavior of the online model.

Figure 9 (now Figure 4) now includes suboxic regions at 300 m depth. A sentence is added to the first paragraph of Section 3.1 (page 6):

While the global annual average rate of denitrification is higher in the online model ($5.63 \times 10^{-13}$ mol N m$^{-3}$ s$^{-1}$) than in the offline model ($5.55 \times 10^{-13}$ mol N m$^{-3}$ s$^{-1}$), the offline model has more grid points with very high values (not shown).

Page 6, line 15, and Figures 10 and 11: "Polar filtering" is blamed for the mismatch at high latitudes. Seems likely that this is not the cause. The mismatch is not really in the "polar" regions - is this filtering really applied at 50-60oS in the ACC? Also, the nutrients are too high in the surface S. Ocean and too low in the deep S. Ocean — this seems to implicate the biological pump (e.g. particle formation/sinking) as the culprit. A more careful discussion of these differences and their possible causes is warranted.

Figures 10 and 11 are now Figures 5 and 6. The identical biogeochemical code is used in the online and offline models- particle formation and sinking is the same. The only way the biological pump could vary between models is by differences in the biomass at the surface. Difference plots have been added to Figure 2 which show generally higher biomass in the offline model, which would increase particle export, lowering surface nutrients and raising them in the deep ocean. This is consistent with the surface difference plot of phosphate in the Southern Ocean, which shows the online model has higher surface phosphate (lower primary production). The deep Pacific and deep South Atlantic also shows the offline model has higher phosphate which is also consistent with higher primary production in the offline model Southern Ocean.

The drivers of the differences in primary production must be due either to differences in the application of external forcing or to physical differences arising from the method of integration. Surface processes in the polar regions are likely to have downstream affects, which is why we suspect the absence of polar filtering may be contributing to differences at 50-60 degrees. Of course it could also be slightly different application of external forcing has an impact on the biology equations. However, in the Arctic, differences in deep alkalinity exist even at 60N even though biomass does not show a clear bias in the offline model. Polar filtering is the only plausible explanation for this.

The phosphate and nitrate paragraph in Section 3.1 has been amended to reflect the possibility seasonal forcing may also be driving differences in phytoplankton biomass at the high latitudes:

The absence of polar filtering in the offline model is contributing to these differences, as is slightly higher primary production in the offline Southern Ocean which enhances the biological pump (Fig. 2). This higher primary production may be due to slight differences in the application of external forcing, which is expected to introduce some bias to regions with strong seasonality. All of the above biases are small relative to discrepancies between the models and observations (see below).

Section 3.2.2 "Seasonal cycle". More discussion of how the seasonal cycle is handled is needed. Equations are needed. Is there a separate TM for each month? And then Euler forward (or backward) is applied to time-step the model? I'm assuming this is the case, but this should be made explicit. In regards to the difference between the TMM and online model, for example as seen in the Indian Ocean for phosphate (Fig. 14), how much is due to the time-averaging of the TMM, and how much to the time- stepping scheme? Would some of these differences be reduced with a more robust time-stepping scheme? e.g. Adams-Bashforth or Crank-Nicholson.

In response to this and comments by the other reviewers we have greatly expanded Sec. 2.1 and added a paragraph to Sec. 2.2, which we hope provides the additional details that the referee is asking for.

As for the differences between online and offline runs, they could arise from a number of factors, including time averaging of circulation (TMs) and other forcing fields (see last paragraph of Sec. 2.2) and differences in the time-stepping scheme. Unfortunately, short of implementing the same scheme offline as used by UVic, it is difficult to pinpoint the exact reason. As stated on Page 4, line 5, one of the key advantages of the TMM, as we see it, is that the same underlying framework can be used to "mix and match" circulations and biogeochemical models. This would be lost – or at least become quite complicated – if we had to implement time-stepping schemes for every ocean model from which TMs have been extracted (not only UVic, but others such as MITgcm and NEMO). Obviously this means that even without time averaging the offline run may not always exactly replicate the online one.

---

## Author Response (AR2)

Dear Dr. Halloran,

We have clarified the first point as suggested. The second point is still causing us some confusion because the online GBC model is embedded in a "coupled" climate model, but one which does not prognostically simulate things like wind speed. We have attempted to clarify the point as we understand it on page 4, lines 24-25 and page 7, lines 16-17.

Furthermore we checked the URLs given in the code availability section and found one was incorrect, and have modified the link accordingly.

Best regards,
Karin